# Differential Diagnosis of Pigmented Lesions in the Oral Mucosa: A Clinical Based Overview and Narrative Review

**DOI:** 10.3390/cancers16132487

**Published:** 2024-07-08

**Authors:** Silvio Abati, Giacomo Francesco Sandri, Leonardo Finotello, Elisabetta Polizzi

**Affiliations:** 1Clinical Unit of Oral Medicine and Pathology, Dental School, IRCCS San Raffaele Hospital and University Vita-Salute San Raffaele, 20132 Milan, Italy; giacomofrancescosandri@gmail.com (G.F.S.); finotelloleonardo@gmail.com (L.F.); 2Center for Oral Hygiene and Prevention, Dental School, IRCCS San Raffaele Hospital and University Vita-Salute San Raffaele, 20132 Milan, Italy; polizzi.elisabetta@hsr.it

**Keywords:** oral health, oral medicine, oral pathology, pigmented lesions, malignant melanoma, oral pigmentation, melanin, mouth mucosa

## Abstract

**Simple Summary:**

This paper addresses the complex diagnostic challenges of pigmented lesions in the oral mucosa, ranging from benign to potentially malignant conditions. Our study provides a narrative review with a clinical case overview to guide clinicians in differentiating these lesions. By classifying pigmented lesions based on clinical and histological features, we emphasize the need for a structured approach to diagnosis. The study incorporates a retrospective analysis of cases from our oral medicine experience, utilizing clinical pictures and relevant histology. Our findings aim to enhance early and precise diagnosis, improving patient management and outcomes in dental and medical practice.

**Abstract:**

This paper examines the clinical differentiation of pigmented lesions in the oral mucosa, which poses significant diagnostic challenges across dental and medical disciplines due to their spectrum from benign to potentially malignant conditions. Through a literature review and analysis of clinical cases, this study clarifies current diagnostic methodologies, with an emphasis on differential diagnosis, to provide a practical guide for clinicians. The classification of pigmented lesions, such as endogenous, focal melanocytic, and multifocal pigmentation, based on clinical and histological features, highlights the necessity for a structured and informed approach. A retrospective examination of cases from our oral medicine and pathology clinic, coupled with analysis of photographic and histological records, aids in classifying these lesions. This fosters a better understanding and promotes informed discussions among clinicians, ultimately aiming to enhance early and precise diagnosis, thus improving patient management and outcomes.

## 1. Introduction

The human oral mucosa displays a vast spectrum of colors, ranging from pale coral pink to yellow, red, and brown, extending beyond mere physiological variations to encompass an array of pathological states. This diversity in hue and tone is influenced by several factors, notably the degree of keratinization, the density and melanogenic activity of melanocytes within the basal epithelial layer, the extent of vascularization, and the intrinsic properties of the submucosal tissue, which may include muscle, bone, or cartilage. In this context, the inherent coloration of the oral mucosa in individuals with lighter skin tones may vary from shades of white to deep red-purple, while in those with darker complexions, the gums, buccal mucosa, and labial tissue often exhibit a range of brown shades. This chromatic variability not only reflects the genetic factors that regulate melanin synthesis by melanocytes and its subsequent transfer to adjacent keratinocytes but is also influenced by external factors such as trauma, inflammation, hormonal fluctuations, exposure to certain pharmacological agents, external agents that can pigment the mucosa, and radiation, all of which can induce an increase in melanin production.

The clinical landscape of pigmented lesions within the oral mucosa is vast and complex, necessitating a rigorous and nuanced approach to differential diagnosis. These lesions, ranging from benign melanotic macules and physiological pigmentation to melanomas and indicators of systemic diseases, pose formidable diagnostic challenges. Accurate evaluation of these lesions is critical, as it spans a diagnostic continuum that includes benign conditions such as freckles (ephelides), oral melanotic macules, and smoker’s melanosis, to more severe concerns like oral malignant melanoma and pigmentation indicative of systemic conditions such as Addison’s disease and Peutz–Jeghers Syndrome. Moreover, the evaluation of pigmented lesions in the oral cavity is further complicated by multifactorial etiologies, including endogenous factors like nevi and exogenous causes such as amalgam tattoos, making accurate diagnosis and management of these lesions pivotal in clinical practice.

The identification and appropriate management of these conditions underscore the intersection of oral medicine and dermatology. It necessitates a comprehensive understanding of the physiological variations in oral mucosa coloration, alongside an ability to discern pathological states from normal conditions.

## 2. Melanin-Associated Pigmented Lesions

Melanin-associated pigmented lesions in the oral cavity encompass a wide range of conditions, from benign entities such as ephelides (freckles) and melanotic macules, to melanomas. These lesions, characterized by increased deposition of melanin in the oral mucosa, present a diagnostic challenge due to their varied etiology and clinical presentation. A thorough understanding of these melanin-rich lesions is crucial for clinicians to accurately differentiate between benign and non-benign conditions, ensuring appropriate management and patient care. The ability to distinguish these lesions based on their clinical and histopathological features is essential for effective diagnosis and underscores the importance of a comprehensive examination and, when necessary, biopsy and histological analysis.

### 2.1. Ephelides (Freckles)

Ephelides, commonly referred to as freckles, are benign, hyperpigmented entities characterized by augmented melanin synthesis and deposition, without a proportional increase in melanocyte density [1]. They predominantly appear in sun-exposed dermal areas in individuals with lighter skin tones and can also manifest on the vermilion border of the lips (Figure 1), presenting unique diagnostic challenges [2].

The genesis of ephelides is intricately linked to genetic predispositions, notably associated with polymorphisms in the melanocortin-1 receptor (MC1R) gene, a key regulator of melanin biosynthesis [3]. These genetic determinants, in conjunction with environmental catalysts such as ultraviolet (UV) radiation exposure, precipitate localized melanin overproduction characteristic of ephelides [4]. Unlike other melanin-rich lesions, ephelides tend to intensify under sun exposure and fade in their absence, illustrating the dynamic modulation of melanin synthesis in response to UV radiation [5].

The management protocol for oral ephelides primarily focuses on vigilant monitoring for changes in dimensions, pigmentation, or morphology indicative of an alternate pathology. Given their benign nature, therapeutic intervention is generally unwarranted [1]. Nonetheless, educating patients about the importance of UV protection is paramount to mitigate the aggravation of existing lesions and the emergence of new ones, particularly in individuals predisposed to skin cancers due to substantial sun exposure or genetic susceptibilities [5].

### 2.2. Oral/Labial Melanotic Macules

In the field of oral medicine, oral melanotic macules are recognized as a prevalent category of benign pigmented lesions, characterized by a localized increase in melanin concentration [6]. This enhanced melanin deposition occurs without a corresponding proliferation of melanocytes, distinguishing these macules from other pigmented oral pathologies. Clinically, these macules range in hues from light brown to a more pronounced black, primarily dictated by the depth and density of melanin within the epithelial and, occasionally, subepithelial layers (Figure 2) [7].

The etiology of these macules is multifactorial, combining idiopathic elements with a spectrum of systemic diseases, hormonal fluctuations, and specific incidents of localized trauma. The precise mechanisms leading to the heightened activity of melanocytes in melanin production remain partially understood. It is speculated to involve a complex interplay of both intrinsic genetic factors and extrinsic environmental influences, such as exposure to ultraviolet light, which is known to trigger melanin production as a protective response [8]. Additionally, various medications and systemic illnesses have been associated with the emergence of oral melanotic macules, indicating a possible correlation between systemic health and concentrated oral pigmentation phenomena [6].

Clinically, these macules are commonly situated on the lower lip. However, their presence is not restricted to this region alone; presentations on the buccal mucosa, gingiva, hard and soft palates, and the dorsal surface of the tongue have been documented (Figure 3). These macules, typically isolated in appearance, vary in size, often occupying a few millimeters in diameter. Despite their benign character, the identification of oral melanotic macules warrants a comprehensive clinical assessment [9].

The clinical evaluation of oral melanotic macules typically relies on comprehensive visual inspection and an in-depth review of the patient’s medical history. However, atypical features or documented changes in the lesion’s appearance necessitate additional diagnostic measures [10]. In this case, histological assessment through biopsy becomes essential. Histopathological findings often reveal increased melanin deposition in the epithelial basal layer and may include melanin-laden macrophages within the connective tissue, without a corresponding rise in the number of melanocytes [9].

Management of oral melanotic macules is generally conservative, prioritizing ongoing surveillance to monitor for any alterations indicative of malignant transformation or other clinical concerns [11].

### 2.3. Racial Pigmentation

Racial or ethnic physiological pigmentation refers to the alteration of the typical coral pink pigmentation of the oral mucosa due to increased cellular activity of melanocytes [12,13], notably an augmented quantity of melanin deposited on the epithelial basal layer [8,14]. This form of pigmentation is the most commonly observed in the oral mucosa, representing 39.9% of all oral pigmentations [15]. It predominantly affects individuals with dark skin tones without gender predisposition and is most frequently diagnosed in the first two decades of life [16,17]. Oral ethnic pigmentation has been documented in 13.5% of children in Israel [18].

Pigmentation manifests as dark brown to black in color, can intensify with age, and is influenced by systemic therapies, smoking, and hormones [19,20]. The most affected site is the gingiva, particularly the attached gingiva, excluding the marginal border [21,22], in 72% of cases of ethnic pigmentation [15] other affected sites include the buccal mucosa, lips, palate, and tongue (Figure 4) [16,19]. Microscopically, there is no observed increase in the number of melanocyte cells or their migration, but rather an increased deposition of melanin on the epithelial basal layer and lamina propria.

This type of pigmentation is included in the differential diagnosis with some systemic syndromes, including Peutz–Jeghers syndrome, Addison’s disease, and drug-induced pigmentation. The diagnosis is based on oral clinical examination, and since it is asymptomatic and innocuous, ethnic pigmentation does not require any treatment except for aesthetic reasons [13,20].

### 2.4. Oral Melanoma

Oral melanoma is a rare malignancy originating from the malignant transformation of melanocytes within the oral cavity. Unlike the more common cutaneous melanomas, oral melanomas exhibit distinct features and challenges. Primary oral melanoma accounts for approximately 1% of all melanomas [23,24] and 0.26–0.5% of all malignant neoplasms in the oral cavity [25,26]. Increased incidence rates have been noted in dark-skinned populations, particularly in Uganda [27]. Gender predisposition remains unclear; however, some studies indicate a male-to-female ratio of 2.5–3:1. [25], while others report a female predominance with a ratio of 1:1.7 [28]. Consensus exists regarding the typical onset age in the sixth and seventh decades of life [25,28,29]. It is a subset of mucosal melanomas, a rare but aggressive form of melanoma occurring in the head and neck regions [30]. These diseases often result in diagnostic delays, leading to a low outlook for patients. The 5-year survival rate is approximately 25% [31].

Another form of oral melanoma is oral amelotic melanoma, which lacks pigmentation, with histological analysis showing melanin in less than 5% of tumor cells. Though amelanotic melanomas are uncommon, representing less than 2% of all melanoma cases, they constitute 40% up to 75% of melanomas found in the oral cavity [32]. Oral amelanotic melanoma predominantly affects the maxillary gingiva, followed by the palate, and infrequently involves the mandibular gingiva [33]. The clinical lack of pigmentation poses a diagnostic challenge for this type of melanoma. It appears as asymptomatic, irregular, erythematous, flat, or nodular, lesions and It is more likely to involve regional lymph nodes and metastasize to distant sites, particularly the lungs [34].

The etiology of oral melanoma remains largely unclear: unlike the more common cutaneous melanomas, which are primarily caused by the sun’s ultraviolet (UV) rays in 86% of cases, some researchers suggest that tobacco use and chronic irritation from malocclusion may play a role in its development. [35,36]. In approximately 30% of cases, the onset of the disease is preceded by a benign pigmented lesion [37,38], though it typically develops de novo on seemingly normal oral mucosa. The initial signs and symptoms often include a lesion that is soft to palpation, usually pigmented, initially asymptomatic, with irregular borders and a color spectrum ranging from dark brown to bluish-black, occasionally presenting variations of gray, purple, and red due to ulcerations, or depigmentation (Figure 5) [39,40]. The primary sites of affliction include the mucosa of the hard palate, followed by the maxillary gingiva and alveolar mucosa [29,41].

Unlike its cutaneous counterpart, oral melanoma lacks histopathological parameters that can reliably predict prognosis, for example, the Clark and Breslow classification and the ABCDEFG diagnostic scale that is normally used to diagnose nevi on the skin is not used for the pigmented lesions of the mouth mucosa. Additionally, the depth of infiltration does not correlate with disease outcome [42].

The prognosis is dire, with a 5-year survival rate between 10 and 20% and a median survival of 25 months. The 10-year survival rate is reported to be 0% [24,27,43], reflecting the challenge of achieving wide surgical margins due to anatomical constraints. Many patients develop widespread metastases affecting the lungs, liver, brain, bone, and lymph nodes. Accurate diagnosis typically requires an incisional or excisional biopsy for histological examination. Immunohistochemical biomarkers, including S-100 (90% positivity), HMB-45 (95.5% positivity), and Melan-A (100% positivity), play a fundamental role in confirming the diagnosis. The World Health Organization (WHO) categorizes melanoma into four primary types: superficial spreading, nodular, acral lentiginous, and lentigo maligna [28,44]. The two most common growth patterns of oral melanoma are (i) a *superficial spreading radial growth pattern*, featuring pagetoid spread with large melanoma cells in the superficial epithelium, and (ii) a less common but highly aggressive *nodular growth pattern* that often infiltrates deeply into the skin and presents as a dark, raised bump [45]. The TNM staging system was updated by the American Joint Committee on Cancer in 2017 with the 8th edition of the Cancer Staging Manual, excluding the possibility of having stage I and II disease. Stages are classified as follows: stage III, tumors limited to the mucosa and immediately underlying soft tissue; stage IVa, a moderately advanced local disease involving deep soft tissue, cartilage, bone, or skin; and stage IVb, an advanced local disease involving the brain, dura, skull base, lower cranial nerves, and masticatory space [46,47]

Proper imaging, including MRI and PET/CT scans, is crucial for assessing the anatomical relationships between healthy tissues and the neoplasm. Surgical excision remains the primary treatment for oral melanoma. Achieving wide and clear excision margins, although challenging, is essential for favorable patient outcomes [41,48].

Adjuvant systemic therapies such as immunotherapy and radiotherapy are also employed in their management. Unfortunately, both have shown limited success in treating advanced-stage melanoma [30,49].

The first-line treatments involve surgical removal with clear margins; although data is limited due to the rarity of these cancers, both radiation therapy and systemic treatments are also employed in their management [31,50].

The management of oral amelanotic melanoma generally includes a combination of surgical intervention and radiotherapy or chemotherapy. Given the rarity, there is a scarcity of data, with only a few case reports and series available. As a result, knowledge about this type of melanoma is still developing, highlighting the need for additional research to more accurately define its clinicopathological characteristics, immune profile, and responses to various treatments [51].

Second-line treatment, such as therapy and immunotherapy for advanced melanoma, including the less common acral and mucosal subtypes, has demonstrated promising results. Traditional chemotherapy using dacarbazine has achieved limited success, although regimens combining albumin-bound paclitaxel and carboplatin appear more effective. Immunotherapy involving immune checkpoint inhibitors (ICIs) such as nivolumab and pembrolizumab has shown reduced efficacy in acral melanoma (AM) compared to cutaneous melanoma. However, targeted therapies aimed at KIT and CDK4/6 mutations have shown potential. The combination of targeted therapies with ICIs or other agents could enhance treatment outcomes [52].

### 2.5. Oral Tobacco Pigmentation/Smoker’s Melanosis

Tobacco-associated oral melanosis is characterized by diffuse, brown to brown-black hyperpigmentation of the oral mucosa, observed in approximately 20–30% of chronic tobacco users (Figure 6). This condition predominantly affects the gingiva, hard palate, and buccal mucosa, with the severity of pigmentation directly proportional to the cumulative exposure to tobacco. It is hypothesized that melanogenesis induced by tobacco consumption acts as a defensive mechanism against the cytotoxic components of tobacco smoke, including polyaromatic hydrocarbons, nicotine, and benzopyrene [53,54]. An investigative study on the Indian population described the prevalence of oral melanin pigmentation among both smoke and smokeless tobacco users, thereby broadening the understanding of tobacco’s impact on oral health beyond conventional smoking practices [55]. Notably, cessation of tobacco use is associated with a gradual reduction in melanotic manifestations, highlighting the reversible nature of smoker’s melanosis upon withdrawal [56].

Despite its benign pathology, differential diagnosis of smoker’s melanosis is imperative to distinguish it from other melanin-rich oral lesions such as idiopathic melanotic macules, inflammatory melanosis, and medication-induced pigmentation, which share clinical and histopathological features [57]. Importantly, smoker’s melanosis lacks malignant potential, distinguishing it from other pigmented oral pathologies with neoplastic implications. Moreover, the systemic consequences of tobacco use are profound, with epidemiological evidence demonstrating a correlative increase in colorectal cancer risk among tobacco users, thereby highlighting the extensive adverse health outcomes associated with tobacco consumption [58]. Clinically, smoker’s melanosis appears as widespread, uneven darkening, mainly impacting the anterior vestibular mucosa. The degree of this pigmentation is closely linked to how long and how intensely the individual has smoked. [53].

This condition underscores the importance of a comprehensive medical and social history, including tobacco use, in the differential diagnosis of oral pigmentation [57]. The reversibility of smoker’s melanosis upon cessation of smoking further emphasizes its direct link to tobacco exposure [55].

### 2.6. Melanoacanthoma

Oral melanoacanthoma is recognized as a non-neoplastic, benign lesion characterized predominantly by excessive proliferation of dendritic melanocytes within the epithelial layer of the oral mucosa, presenting a clinical and histopathological profile distinct from other pigmented oral lesions [59]. Typically manifesting as solitary, well-circumscribed macules or plaques, these lesions exhibit brown to black pigmentation and show a marked predilection for the buccal mucosa, although they may also affect the lips, palate, and gingiva [60].

The sudden appearance and rapid expansion of oral melanoacanthoma highlight the necessity of differential diagnosis to distinguish it from other malignant conditions, such as melanoma and various pigmented pathologies. Histopathological examination typically reveals a notable increase in dendritic melanocytes throughout the spinous layer, without evidence of cellular atypia [61]. In such cases, an incisional biopsy is mandatory. Following diagnosis, treatment is generally not indicated [7].

## 3. Non-Melanin Associated Pigmented Lesions

### 3.1. Endogenous Pigmentation

Oral pigmented lesions caused by endogenous pigments, particularly those derived from blood, represent a significant clinical concern. These lesions can be classified based on their origin and the nature of the pigments involved. For blood pigments, the coloration is primarily due to the presence of hemoglobin, hemosiderin, and other hemoglobin degradation products, which impart a characteristic red, bluish-purple, brown, or black color to the lesions [2].

Blood extravasation into oral tissues can lead to various forms of pigmented lesions, such as hematomas, petechiae, purpura, and ecchymoses. These manifestations result from the accumulation and subsequent degradation of hemoglobin into bilirubin and biliverdin. Common traumatic events in the oral cavity that may cause blood extravasation and subsequent hyperpigmentation include bites and iatrogenic procedures such as dental interventions [62].

The diagnosis of these lesions is primarily based on clinical examination, which can be implemented with the clinical test of diascopy (vitroscopy). This procedure differentiates pigmented from vascular lesions by removing the camouflaging effect of congested blood to reveal the true color of underlying lesions. It involves pressing a glass or plastic diascopy device, usually a microscope slide, against a cutaneous or mucous lesion. The characteristic blanching effect occurs due to blood dissipating intravascularly under compression, giving the tissue a pale appearance [63].

Patients with hemochromatosis, a condition also known as “bronze diabetes”, often present with a grayish-blue pigmentation of the hard palate, gingiva, and buccal mucosa [64]. This pigmentation results from the accumulation of iron-containing pigments, (ferritin and hemosiderin) within the skin and mucous membranes. This iron accumulation results from a defect in iron metabolism, leading to excessive absorption and deposition in body tissues [65].

In patients with beta-thalassemia, diffuse black-brown pigmentation can be observed. This inherited blood disorder results in abnormal hemoglobin production and commonly presents at the junction between the hard and soft palate (Figure 7). The pigmentation is due to the increased deposition of hemosiderin, a degradation product of hemoglobin. Beta-thalassemia causes accelerated red blood cell destruction, resulting in the release and accumulation of hemosiderin in the tissues [66,67].

Oral varices (OV) are one of the most frequent changes in the oral mucosa, typically found under the tongue and in the lower lips in elderly patients. They are benign, painless, blue to purple dilatations of veins and/or venules with a nodular pattern. As they become more prominent with age, they can interfere with mastication. The pathogenesis is not clear; regular follow-ups are necessary in order to monitor this condition [68].

### 3.2. Exogenous Pigmentation and Amalgam Tattoos

Exogenous oral pigmentation is a noteworthy subject due to its frequent occurrence and the diverse sources from which it arises. Typically benign, this pigmentation results from foreign substances introduced into the oral mucosa [69]. A common example is the amalgam tattoo, which occurs when small particles from dental amalgam fillings become accidentally embedded in the soft tissues of the mouth during dental procedures (Figure 8). These tattoos appear as gray, blue, or black spots, usually near the site of dental work [70]. Amalgam tattoos are a prevalent form of exogenous pigmentation, arising typically from dental procedures where amalgam material is inadvertently embedded in the soft tissues. They are characterized by their distinctive blue or black coloration and are identified near areas of dental restorations [71]. Exogenous pigmentations can also occur from old materials used in endodontic fillings, which can traverse the bone and tissues, causing pigmentation of the mucosa at the level of the involved tooth apex. Diagnosis primarily relies on clinical examination, but radiographs or biopsy should be employed when the origin of the pigmentation is unclear, to distinguish them from more serious conditions. Radiographic examination is preferred for the differential diagnosis of amalgam/metal tattoos when suspected. A biopsy is recommended when radiographic examination does not reveal the presence of metal particles, even if clinical suspicion leans towards an amalgam tattoo, particularly in solitary cases, since these lesions enter into differential diagnosis with nevi and melanoma [72]. Management of these pigmented lesions generally does not require intervention unless there is an aesthetic concern or a need to confirm the benign nature of the pigmentation. The prognosis for patients with amalgam tattoos and similar forms of exogenous pigmentation is generally good, underscoring the importance of awareness and understanding of these conditions among dental professionals [73].

## 4. Drug Induced Oral Melanosis

### Drug-Induced Mucosal Pigmentation

Drug-induced oral melanosis results from the chromatic alteration of the oral mucosa, with an estimated 10–20% of acquired oral pigmentations induced by certain medications. Antimalarials such as chloroquine, hydroxychloroquine, and quinacrine are prominently implicated in the development of drug-induced oral pigmentations [74]. Additional medications involved include antipsychotics such as chlorpromazine; chemotherapeutics like bleomycin, imatinib, busulfan, fluorouracil; antifungal drugs including ketoconazole; antimicrobial drugs such as tetracycline and minocycline; and antiretroviral drugs such as zidovudine [7,75,76]. Pigmentation may affect all oral surfaces, be localized to a single site, or spread to multiple sites, with the hard palate, gingiva, and tongue being the most affected (Figure 9).

At the pathophysiological level, two mechanisms underlie the formation of these pigmentations: (i) directly or indirectly inducing the synthesis and release of melanin by melanocytes, resulting in true pigmentation; (ii) through the deposition of drug precipitates in the submucosa or direct binding of the drug to melanin thus imparting a blue to black coloration to the oral mucosa [7]. The specific mechanism by which melanin synthesis is increased is not fully understood, though it is possible that metabolites of some drugs may stimulate melanogenesis. Microscopically, hyperpigmentation and melanin leakage are observed with no increase in the melanocyte numbers. Clinical diagnosis is simplified if a temporal association is established between drug intake and the onset of pigmentation; otherwise, a diagnostic biopsy is recommended. In most cases, the discoloration tends to fade within a few months after the drug is discontinued [74]. Malignant transformation of these lesions has not been reported [17].

## 5. Systemic Disorders

### 5.1. Peutz–Jeghers Syndrome

Peutz–Jeghers Syndrome (PJS) is a hereditary autosomal dominant disorder characterized by the formation of unique polyps in the gastrointestinal tract and specific mucocutaneous pigmentation. Mutations in the STK11 gene on chromosome 19p13.3 are strongly linked to this syndrome. Those with PJS are at a much higher risk of developing several types of cancer, especially in the breast and gastrointestinal regions [77]. The rarity of PJS poses challenges in establishing evidence-based surveillance and management protocols. However, the primary focus of early endoscopic surveillance is to identify polyps that could lead to intussusception or obstruction rather than for cancer detection, with cancer surveillance becoming a critical component of management as patients age [78].

The diagnosis of Peutz–Jeghers Syndrome (PJS) can be made if a person has two or more histologically confirmed Peutz–Jeghers polyps, any Peutz–Jeghers polyps along with a family history of the syndrome, or the typical mucocutaneous pigmentation in the presence of a family history of PJS [77]. The syndrome has a higher risk of malignancy compared to the general population, including heightened risks for gastrointestinal, breast, lung, and genitourinary cancers [79].

The oral manifestations, particularly the pigmented lesions on the lips and inside the mouth, serve as crucial diagnostic features and visual markers that can alert healthcare providers to the potential presence of PJS [80]. While these lesions are benign and highly indicative of the syndrome, differential diagnosis is essential, as oral pigmentation can also be associated with melanotic macules, nevi, and ephelides on the lips (Figure 10). Accurately identifying the pigmented lesions of PJS and differentiating them from conditions such as melanotic macules, nevi, and ephelides is crucial for guiding appropriate clinical management and surveillance [79].

### 5.2. Addison’s Disease

Addison’s disease, also known as adrenal insufficiency or primary hypoadrenalism, is an endocrinopathy characterized by reduced glucocorticoid production due to progressive destruction of the adrenal cortex, typically caused by autoimmune disease, cancer, or infection [81]. The diminished production and subsequent release of adrenocorticoid hormones in the blood lead to dysfunction of the hypothalamus-pituitary-adrenal gland axis, particularly resulting in elevated levels of adrenocorticotropic hormone (ACTH). This increased ACTH production stimulates melanocyte-stimulating hormone (α-, β-, and γ-melanotropins), thereby causing diffuse pigmentation on both cutaneous and oral surfaces (Figure 11).

Cutaneo-mucosal hyperpigmentation affects 92% of Addison’s disease patients [82]. These oral pigmented lesions present as diffuse brown macules that can affect all oral surfaces, typically appearing and progressing in adult life [16]. Diagnosis of these oral pigmented lesions is primarily based on the patient’s clinical history. Oral pigmented lesions caused by Addison’s disease do not require any treatment.

## 6. Post-Inflammatory Pigmentations

### Pigmented Oral Lichen Planus

Oral lichen planus (OLP) is a persistent inflammatory disorder that targets the mucous membranes within the mouth. It manifests as white, lace-like patches or red, inflamed tissues, often leading to considerable discomfort, such as pain and a burning feeling. The precise cause of OLP is not fully understood, but it is thought to involve an immune-mediated process, where the immune system erroneously attacks the cells of the mucous membranes. [83].

Among the various forms of lichen planus, there exists a distinct pigmented variant known as lichen planus pigmentosus (LPP). This variant was first identified in India by Bhutani et al. in 1974, who introduced the term to describe its unique presentation. LPP is distinguished by its slow onset and the appearance of small, dark brown or black macules that typically emerge on sun-exposed areas of the skin (Figure 12). Over time, these macules can merge to form larger patches of hyperpigmentation. Although LPP primarily impacts the face, torso, and upper limbs, LPP rarely involves the oral mucosa. Notably, the condition does not affect the palms, soles, or nails [84,85].

Histologically, LPP exhibits notable atrophic alterations in the epidermis, vacuolar degeneration of the basal cell layer, and melanin incontinence within the dermis. This is further accompanied by the presence of dispersed melanophages and a sparse infiltrate around hair follicles or blood vessels. This pigmented variant shows considerable histopathological overlap with erythema dyschromicum persistant, though differences in clinical presentation and immunologic features suggest they may be distinct entities. Despite this, some dermatologists debate whether these conditions should be considered separate diseases [86,87].

Patients with LPP have been found to have associations with several systemic conditions, such as hepatitis C virus infection, frontal fibrosing alopecia, Bazex syndrome, and nephrotic syndrome. An unusual form of LPP, known as LPP inversus, was reported in 2001. Unlike the typical presentation, LPP inversus affects intertriginous areas such as the groins and axillae, predominantly occurring in individuals with lighter skin tones. This variant shares clinical and histopathological features with LPP but presents in areas not exposed to sunlight [88].

## 7. Conclusions

This narrative review comprehensively addresses the complexities involved in the differential diagnosis of pigmented lesions within the oral mucosa. As evidenced through the detailed examination of various case studies and extensive literature review, the accurate differentiation and timely diagnosis of these lesions are crucial due to their diverse etiologic spectrum ranging from benign discolorations to potentially malignant conditions.

In the following Table 1, the clinical features, the causes, the proposed treatments and differential diagnosis of the lesions and conditions described in this review are summarized and compared.

A cornerstone of effective management of pigmented oral lesions is the early and precise identification through rigorous clinical examination and histopathological evaluation via biopsy. The role of the dental hygienist is pivotal, often serving as the first point of clinical contact. Dental hygienists are uniquely positioned to recognize abnormal pigmentation during routine examinations, thereby initiating a referral pathway that leads to specialized care from oral medicine specialists or otolaryngologists.

The differentiation between benign lesions and those with malignant potential cannot be understated, as it directly influences treatment decisions and patient prognosis. Conditions such as melanotic macules, which are generally harmless, and malignant melanomas, which require aggressive management, exemplify the spectrum of clinical presentations encountered in practice. This delineation is supported by a thorough histopathological examination, underscoring the necessity of biopsy in ambiguous cases.

Moreover, the significance of detailed photographic documentation emerges as a valuable tool in monitoring the evolution of pigmented lesions over time, providing a visual record that can be invaluable in tracking changes and responding appropriately.

Comprehensive patient history collection also remains crucial, with the oral medicine specialist integrating data regarding potential systemic diseases, medication use, and lifestyle factors like smoking. This thorough collection of patient information not only aids in crafting a complete diagnostic picture but also enhances the understanding of the systemic implications of oral pigmentation.

In conclusion, the ability to conduct a detailed and accurate differential diagnosis, timely referral to specialists, and the integration of clinical findings with histopathological insights form the cornerstone of managing pigmented lesions in the oral mucosa. Future enhancements in the training of all oral and dental professionals, including dental hygienists, and advances in diagnostic technologies, promise to improve patient outcomes through earlier detection and tailored intervention strategies.

## Figures and Tables

**Figure 1 cancers-16-02487-f001:**
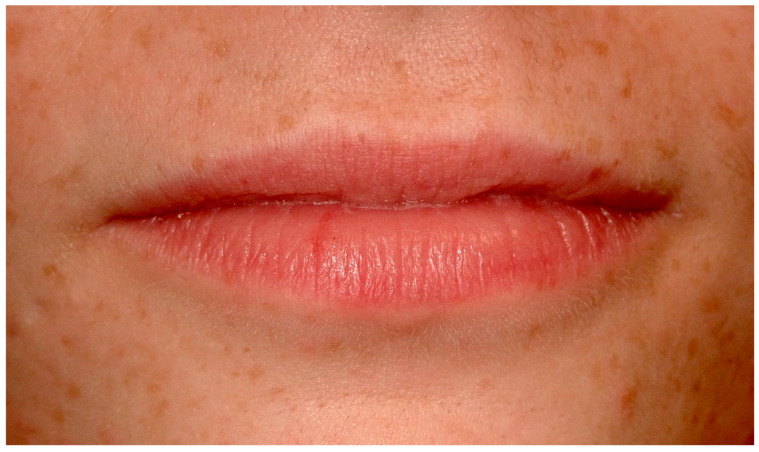
Ephelides on the perioral skin and labial psudomucosae on a 22-year-old patient. Patient refers to these lesions from birth. (archive S.A., patient signed the consent for clinical pictures).

**Figure 2 cancers-16-02487-f002:**
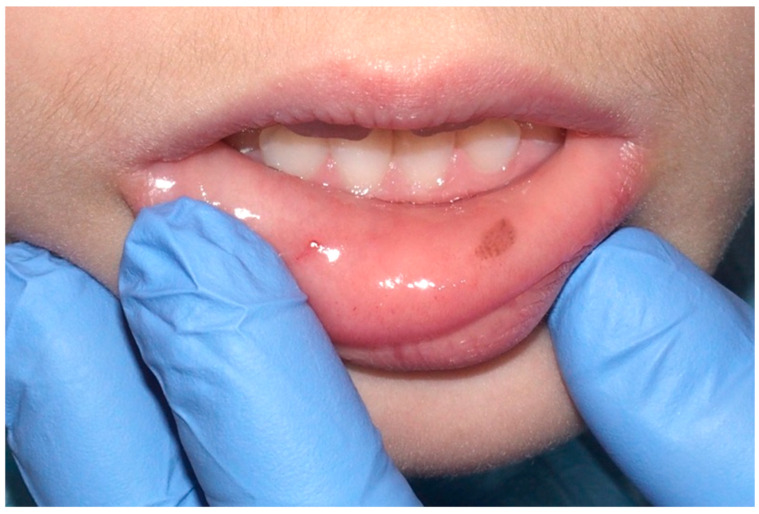
Oral melanotic macula on the lower labial vestibular mucosa on a 7-year-old patient. Parents referred these lesions as being present for over 6 months. (archive S.A., patient signed the consent for clinical pictures).

**Figure 3 cancers-16-02487-f003:**
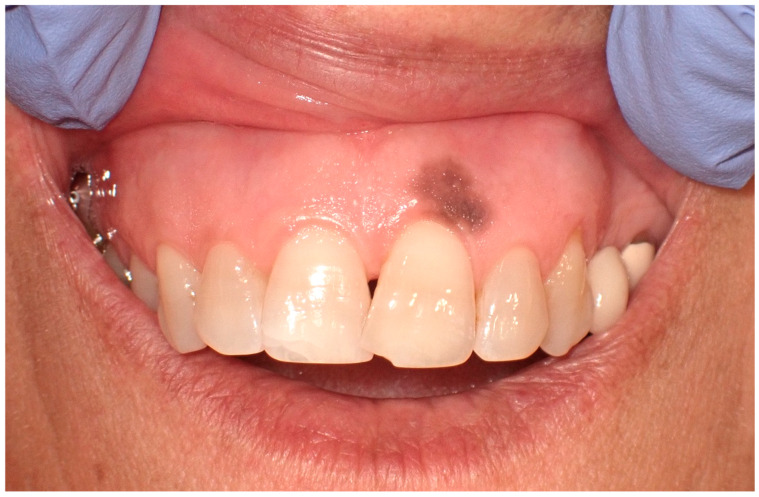
Oral melanotic macula on the upper gingival mucosa on a 40-year-old patient. Patient refers to these lesions as being present since over 6 months. (archive S.A., patient signed the consent for clinical pictures).

**Figure 4 cancers-16-02487-f004:**
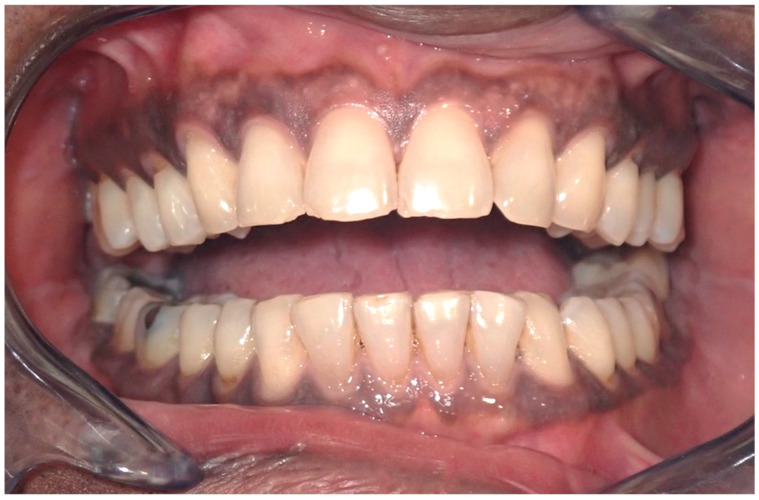
Ethnic pigmentation of the gingival tissues on a 60-year-old patient. Patient refers to these pigmentation from decades. (archive S.A., patient signed the consent for clinical pictures).

**Figure 5 cancers-16-02487-f005:**
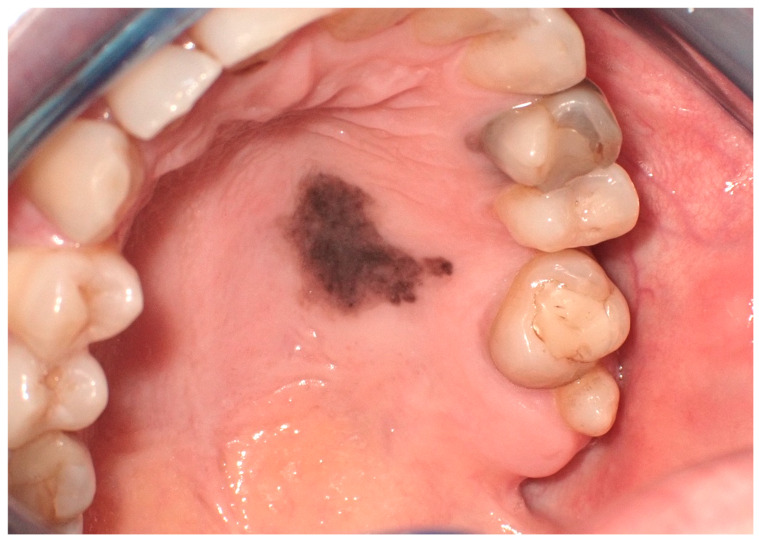
Oral malignant melanoma, in the hard palatal mucosa on a 67-year-old female patient. Patient refers to these lesions as being present for 3 months. (archive S.A., patient signed the consent for clinical pictures).

**Figure 6 cancers-16-02487-f006:**
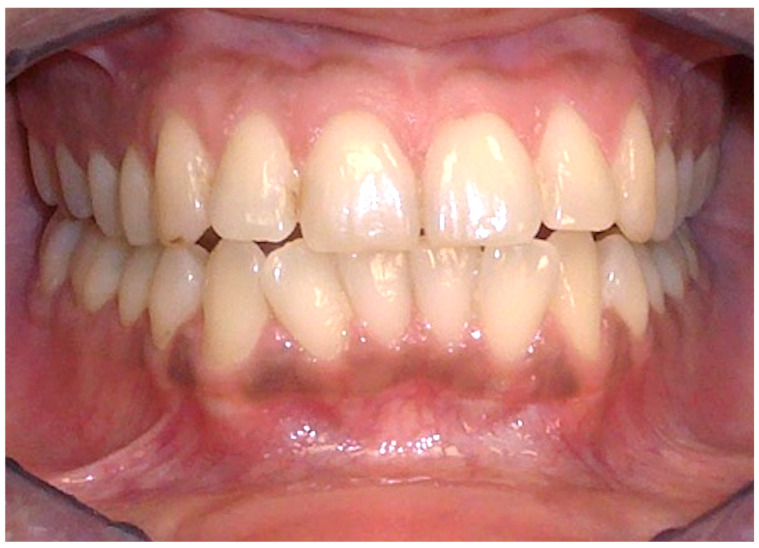
Smoker’s Melanosis in the lower gingival mucosa of a 38-year-old heavy smoker male patient. Patient refers to these lesions as being present for 6 years. (archive S.A., patient signed the consent for clinical pictures).

**Figure 7 cancers-16-02487-f007:**
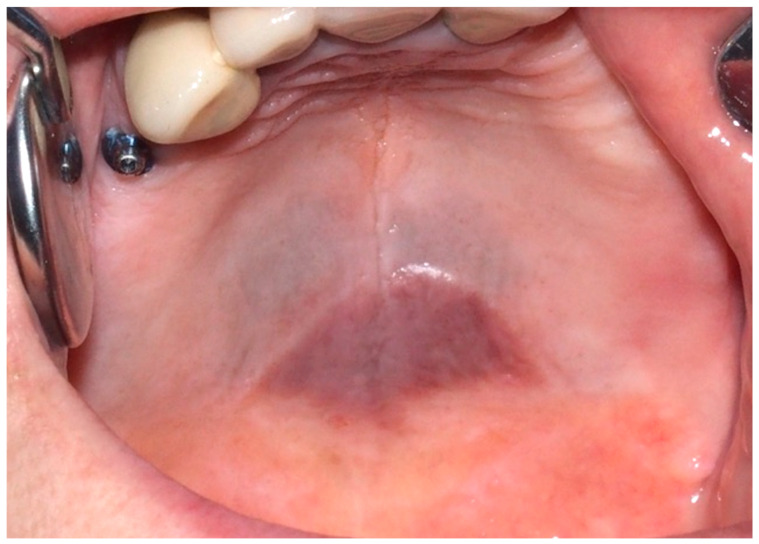
Grayish-blue dark pigmentation at the border between hard and soft palate due to hemosiderin deposition in a 65-year-old female patient affected by beta-thalassemia. Patient refers to these lesions as being present for 10 years. (archive S.A., patient signed the consent for clinical pictures).

**Figure 8 cancers-16-02487-f008:**
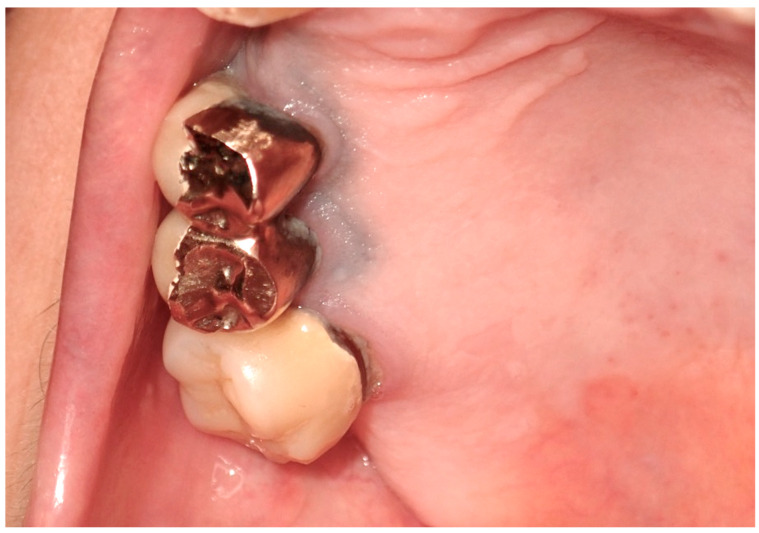
The so-called “amalgam tatoo”, greyish pigmentation of the marginal gingival from particles of an amalgam of the previous dental restoration of upper premolar teeth sprayed during the preparation of the prosthetic abutments in a 67-year-old patient. Patient refers to these lesions as being present for 20 years. (archive S.A., patient signed the consent for clinical pictures).

**Figure 9 cancers-16-02487-f009:**
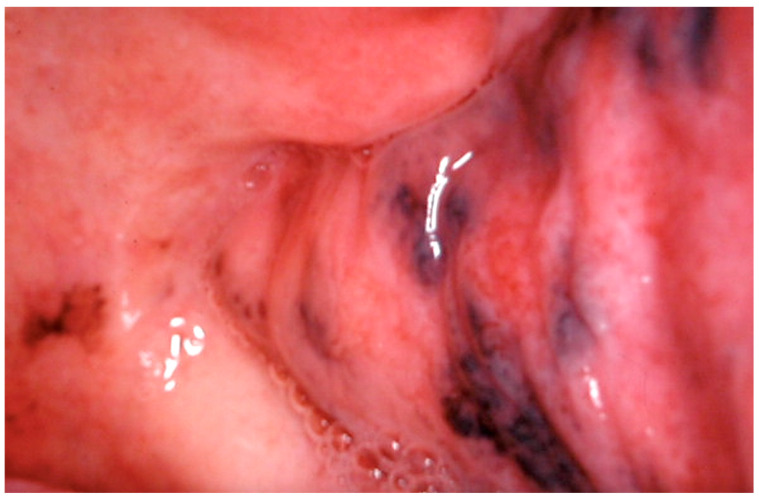
Extensive oral mucosal pigmentation due to long term therapy with systemic hydroxychloroquine in a 70-year-old patient. Patient refer to these lesions as being present for over 10 years. (archive S.A., patient signed the consent for clinical pictures).

**Figure 10 cancers-16-02487-f010:**
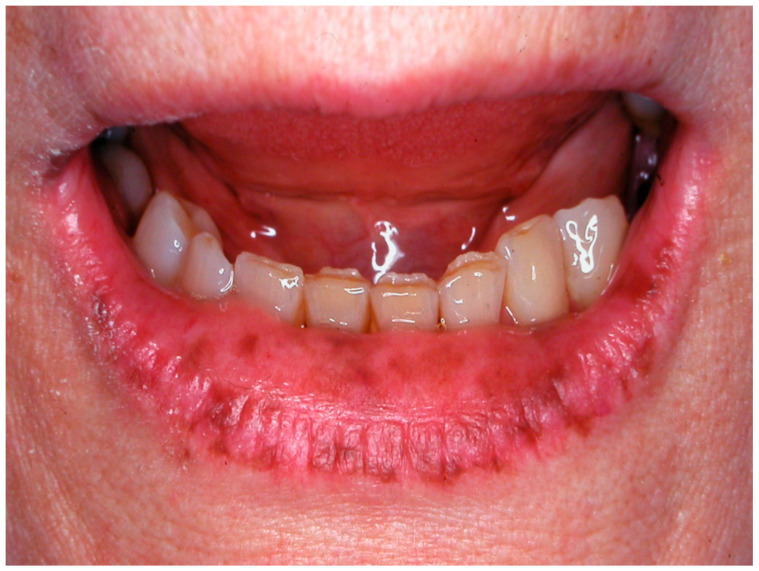
Labial ephelides in a woman affected by Peutz–Jeghers Syndrome in a 63-year-old patient. Patient refer to these lesions as being present for 45 years. (archive S.A., patient signed the consent for clinical pictures).

**Figure 11 cancers-16-02487-f011:**
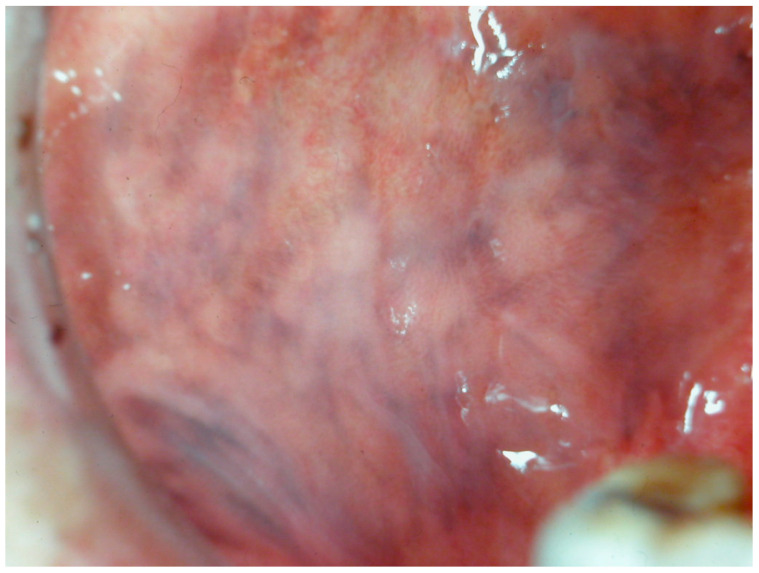
Pigmentation of the cheek mucosa in a patient with Addison’s disease in a 35-year-old female patient. Patient refer to these lesions as being since years. (archive S.A., patient signed the consent for clinical pictures).

**Figure 12 cancers-16-02487-f012:**
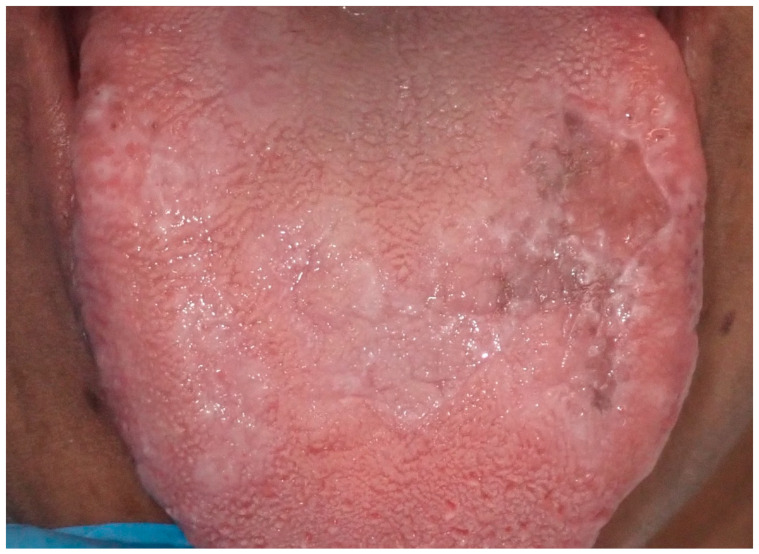
Area of tongue mucosa pigmentation in a patient with reticular oral lichen planus in a 42-year-old male patient. Patient refers to these lesions as being present for several years. (archive S.A., patient signed the consent for clinical pictures).

**Table 1 cancers-16-02487-t001:** Clinical features of pigmented lesions affecting the oral mucosa and differential diagnosis.

Disease	Clinical-Feature	Cause	Treatment	Differential Diagnosis
Ephelides/Freckles	Benign, hyperpigmented pigmentations; caused by an augmented melanin synthesis.	Sun-exposed dermal areas in individuals with lighter skin tones.	Vigilant monitoring for alterations in dimensions; no treatment needed.	MM; PJS.
Oral/Labial Melanotic Macules	Brown-to-black small pigmentations, often a few millimeters. Caused by intrinsic and extrinsic factors.	Multifactorial genetic factors and extrinsic environmental influences. Typically, lower lip.	Surgical biopsy.	Ephelides/Freckles; PJS; OMM;
Ethnic Pigmentation	Predominantly affects dark-skinned individuals; Manifests as dark brown to black; most affected site is the gingiva.	Increased quantity of melanin deposited on the epithelial basal layer.	No treatment needed.	PJS; Addison’s Disease; DIOM.
Oral Melanoma	Brown to black hyperpigmentation of the oral mucosa, often with irregular borders.	Tobacco consumption; Genetical predisposition; Generally unknown.	Incisional/excisional biopsy for the diagnosis and then surgical approach + immunotherapy and radiotherapy.	MM; Melanoacanthoma; Exogenous Pigmentation.
Oral Tobacco Pigmentations	Brown to brown-black hyperpigmentation of the oral mucosa.	Tobacco consumption;	Cessation of tobacco usage.	OMM; Melanoacanthoma.
Melanoacanthoma	Solitary, well-circumscribed macules or plaques, brown to black pigmentation.	Unknown.	No treatment needed.	OMM.
Endogenous Pigmentation	The lesions appear purple, red or brown in color.	Multifactorial; consequence of alterations or genetic disorders.	No treatment needed; Surgical biopsy for differential diagnosis.	OMM; MM; Melanoacanthoma.
Exogenous Pigmentation	Foreign substances being introduced into the oral mucosa; typically gray, blue or black spots; Variable dimensions.	Performed dental works, Can be confirmed by radiograph.	No treatment needed; Surgical biopsy for differential diagnosis.	MM.
Drug-Induced Oral Melanosis	Single or multiple sites (most affected hard palate, gingiva, and tongue); variable in dimensions.	Intake of certain drugs (such as chloroquine, quinacrine);	No oral treatment needed.	MM; Exogenous Pigmentation.
Peutz–Jeghers Syndrome	Pigmented lesions on the lips and inside the mouth, often a few millimeters; Brown-to-black.	Direct consequence of the systemic disease, linked to mutations in the STK11 gene.	No oral treatment needed; Tests for intestinal polyposis can be useful.	Ephelides/Freckles; MM; Exogenous Pigmentation.
Addison’s Disease	Brown hyperpigmentation of the oral mucosa.	Endocrinopathy.	Treat the sistemic condition; Endocrinological tests can be useful.	MM; PJS; Exogenous Pigmentation.
Pigmented Oral Lichen Planus	Brown macules with white patches and stripes.	Post-inflammatory pigmentation on existing OLP.	Vigilant monitoring; no treatment needed.	Oral Tobacco Pigmentations; DIOM; MM.

MM = melanotic macule; PJS = Peutz–Jeghers Syndrome; OMM = oral malignant melanoma; DIOM = drug-induced oral melanosis.

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
