# Peer review of "Differential Diagnosis of Pigmented Lesions in the Oral Mucosa: A Clinical Based Overview and Narrative Review"

_cancers, 2024, doi:10.3390/cancers16132487_

Round 1

Reviewer 1 Report

Comments and Suggestions for Authors

Table 1 should be described under a subheading in the review  instead of the conclusion

Figures can be grouped and figure sources should be acknowledged

Comments on the Quality of English Language

English language is fine

Author Response

REPLIES TO REVIEWER 1

Comment 1: Table 1 should be described under a subheading in the review instead of the conclusion

Response 1: Thank you for pointing out this. We inserted a specific paragraph in the subheadings "Conclusion" describing the scopes and contents of the Table I (lines 540-542)

Comment 2: Figures can be grouped and figure sources should be acknowledged

Response 2: Dear reviewer, we inserted the clinical pictures and the table following the Instructions for Authors reported for the journal Cancers - MDPI ("All Figures, Schemes and Tables should be inserted into the main text close to their first citation and must be numbered following their number of appearance")

Reviewer 2 Report

Comments and Suggestions for Authors

The manuscript by Abati S. et al. investigates pigmented lesions of the oral mucosa, including melanoma, and provides valuable insights for dental surgeons, dermatologists, dental and medical students, and the general public. To my knowledge, this manuscript does not belong in this journal; it needs to delve into the most important lesions, including melanoma.

Revision

Despite the manuscript being interesting, the authors need to review the language; the term "malignant melanoma" is not recommended because there is no such thing as "benign melanoma." The title is not completely associated with the manuscript; it is recommended to be changed.

I don't fully understand the term "potentially malignant melanoma." Please review.

It is necessary to include in all figures age, gender, and time of evolution, for example. Figure 1: "Ephelides on the perioral skin and labial pseudo mucosae." Figure 1: Ephelides on a 13-year-old patient. Patients refer to these lesions from birth. Dr. Abati S. provided the photo, and tutors signed the informed consent.

When describing melanoma, it is necessary to describe types of melanomas, including mucosal melanomas. When describing mucosal melanomas, it is important to describe nodular melanoma and superficial spreading melanoma. It is important to include a clinical description (ABCDEFG). When describing histopathological data, it is necessary to include Clark and Breslow classifications.

In mucosal melanomas, it is important to include WHO classification (WHO, Head and Neck Tumors, 5th ed., 2022).

Amelanotic melanoma is a rare oral mucosa disease; it is necessary to describe it in more detail, as well as its aggression and clinical features.

Figure 5 and all figures Describe the clinical data.

It is necessary to establish an order: first non-neoplastic lesions, second neoplastic lesions, and third malignant lesions.

Comments on the Quality of English Language

It is necessary to review English with an expert; there are terms that are not fully understood.

Author Response

REPLIES TO REVIEWER 2

Comment 1: this manuscript does not belong in this journal.

Response 1: Dear reviewer, we think that in this issue of the journal an update on the generally little known and controversial topic of the pigmented conditions of the oral mucosa could be interesting for readers of different medical backgrounds

Comment 2: the term "malignant melanoma" is not recommended because there is no such thing as "benign melanoma

Response 2: Thank you for this comment. Agree. We have accordingly changed the definitions using the term "melanoma" or "oral melanoma" (lines 187 to 268)

Comment 3: I don't fully understand the term "potentially malignant melanoma."

Response 3: Agree, we addressed your clarification

Comment 4: It is necessary to include in all figures age, gender, and time of evolution,

Response 4: Agree. We included in the legend as you specified

Comment 5: When describing melanoma, it is necessary to describe types of melanomas, including mucosal melanomas. When describing mucosal melanomas, it is important to describe nodular melanoma and superficial spreading melanoma.

Response 5: Agree. We have accordingly done the revisions of the paper (lines 187 to 268)

Comment 6: It is important to include a clinical description (ABCDEFG). When describing histopathological data, it is necessary to include Clark and Breslow classifications. In mucosal melanomas, it is important to include WHO classification (WHO, Head and Neck Tumors, 5th ed., 2022).

Response 6: Agree. We have modified the description including your suggestions; however the ABCDEFG and Clark and Breslow classification are not generally applied and used for oral melanomas

Comment 7: Amelanotic melanoma is a rare oral mucosa disease; it is necessary to describe it in more detail, as well as its aggression and clinical features.

Response 7: Agree. We addressed your suggestion.

Comment 8: Figure 5 and all figures Describe the clinical data.

Response 8: Done in the legends

Comment 9: It is necessary to establish an order: first non-neoplastic lesions, second neoplastic lesions, and third malignant lesions.

Response 9: Dear reviewer, we used a ethiologic and pathogenic classification in the description of the pigmented disease and conditions described

Comment 10: It is necessary to review English with an expert; there are terms that are not fully understood.

Response 10: Thank you for the suggestion. We commissioned the language review of the paper to an expert colleague remained for years in UK and USA

Reviewer 3 Report

Comments and Suggestions for Authors

This paper examines the clinical differentiation of pigmented lesions in the oral mucosa, which poses significant diagnostic challenges.

In Introduction:  please avoid word potentially in formulation" to potentially malignant melanomas"  because  Melanoma is used for the disease itself and  is extremely malignant neoplasm not potentially malignant. Please replace throughout the text (lines 50, 67...) also, do the same in the table.

line 168: "The etiology of oral malignant melanoma remains largely undefined... " according to the literature statistics shows that 86% of melanomas are caused by the sun's ultraviolet (UV) rays, please comment in the etiology.

Line 186: Also, instead of incisional biopsy, which is not recommended due to possibility of hematogenic spread of cancer cell , replace the word with excisional biopsy. Surgeons recommend removing the entire growth when possible and refer to oral/maxillofacial surgeon is mandatory.

Line 303:  "Diagnosis primarily relies on clinical examination, but radiographs or biopsy may be employed..." it is suggested to authors to comment  that radiographic examination is preferred in the differential diagnosis of pigmentation, when amalgam/metal tattoo is suspected. Alternatively, a biopsy is recommended when metal particles cannot be documented radiographically and the differential diagnosis includes nevi or suspicion of melanoma, especially in the case of  solitary lesion.

The reviewer's proposal is to include also in thelist of endogenous pigmentations, in addition to the already mentioned pigmentations originating from blood pigment varix, Kaposi's sarcoma, hereditary hemorrhagic telangiectasia. 

Also it is recommended to mention in diagnosis of pigmented lesions the clinical test diascopy, i.e. vitropressure test for rapid differential diagnosis of vascular from pigmented lesions, which can be of quick help to the clinician in further treatment.

In the table - in endogenous pigmentation, Addison's disease, provide a recommendation for further endocrinological tests, and in the case of Peutz-Jeghers syndrome, tests for intestinal polyposis.

Please mention the source of the clinical pictures. 

Author Response

REPLIES TO REVIEWER 3

Comment 1: Introduction:  please avoid word potentially in formulation "to potentially malignant melanomas"  because  Melanoma is used for the disease itself and  is extremely malignant neoplasm not potentially malignant. Please replace throughout the text (lines 50, 67...) also, do the same in the table.

Response 1: Thank you for your suggestion. You're right. Addressed in Introduction and in Melanoma chapter

Comment 2: line 168: "The etiology of oral malignant melanoma remains largely undefined... " according to the literature statistics shows that 86% of melanomas are caused by the sun's ultraviolet (UV) rays, please comment in the etiology.

Response 2: thank you for this comment. Agree. We have accordingly modified the paper (lines 208-211)

Comment 3: Line 186: Also, instead of incisional biopsy, which is not recommended due to possibility of hematogenic spread of cancer cell , replace the word with excisional biopsy. Surgeons recommend removing the entire growth when possible and refer to oral/maxillofacial surgeon is mandatory.

Response 3: Dear reviewer, you're right, but this is appliable "when possible", for smaller pigmented oral mucosal lesions, under about 2 cm - larger lesions need to be submitted to incisional biopsy to avoid unnecessary mutilation - another case in which incisional biopsy could be a choice is the ulcerated oral melanomas where the invasion of underlying tissues happened

Comment 4: Line 303:  "Diagnosis primarily relies on clinical examination, but radiographs or biopsy may be employed..." it is suggested to authors to comment  that radiographic examination is preferred in the differential diagnosis of pigmentation, when amalgam/metal tattoo is suspected. Alternatively, a biopsy is recommended when metal particles cannot be documented radiographically and the differential diagnosis includes nevi or suspicion of melanoma, especially in the case of  solitary lesion.

Response 4: You're right. Agree. Addressed (line 393)

Comment 5: The reviewer's proposal is to include also in the list of endogenous pigmentations, in addition to the already mentioned pigmentations originating from blood pigment varix, Kaposi's sarcoma, hereditary hemorrhagic telangiectasia.

Response 5: Dear reviewer, you are right regarding the possible dark color of the oral mucosa affected by the vascular disease and conditions that you mention; however they are not pertinent to our topics on the clinics of pigmented lesions

Comment 6: Also it is recommended to mention in diagnosis of pigmented lesions the clinical test diascopy, i.e. vitropressure test for rapid differential diagnosis of vascular from pigmented lesions, which can be of quick help to the clinician in further treatment.

Response 6: Agree, corrected (see line 343)

Comment 7: In the table - in endogenous pigmentation, Addison's disease, provide a recommendation for further endocrinological tests, and in the case of Peutz-Jeghers syndrome, tests for intestinal polyposis.

Response 7: Agree, corrected (see Table I on page 20)

Comment 8: Please mention the source of the clinical pictures. 

Response 8: The clinical pictures included come from the archive of the clinical pictures of patients seen in our Oral medicine and Pathology Dept at the San Raffaele Dental Clinic  - Milano

Reviewer 4 Report

Comments and Suggestions for Authors

The article presents a narrative review on melanocytic lesions of the oral cavity. As a review it is correct and reads easily, having adequate expository clarity. Readers of Cancers, who may not be very familiar with this type of lesions may enjoy this article, although its contribution to the advancement of knowledge is not very important.

Author Response

REPLIES TO REVIEWER 4

Comment to the reviewer

Dear reviewer, thank you for your positive comment on our paper. It could be of interest also for clinician not familiar with oral medicine and pathology

Round 2

Reviewer 2 Report

Comments and Suggestions for Authors

The authors have incorporated all of my suggestions, resulting in significant improvements to the manuscript. Please review malignant melanoma in Figure 5